# Stretchable Carbon and Silver Inks for Wearable Applications

**DOI:** 10.3390/nano11051200

**Published:** 2021-05-01

**Authors:** Andrew Claypole, James Claypole, Liam Kilduff, David Gethin, Tim Claypole

**Affiliations:** 1Welsh Centre for Printing and Coating, Bay Campus, Swansea University, Swansea SA1 8EN, UK; andrew.claypole@swansea.ac.uk (A.C.); j.m.claypole@swansea.ac.uk (J.C.); d.t.gethin@swansea.ac.uk (D.G.); 2Applied Sports, Technology, Exercise and Medicine, Bay Campus, Swansea University, Swansea SA1 8EN, UK; l.kilduff@swansea.ac.uk

**Keywords:** stretchable inks, wearables, carbon ink, graphite nanoplatelet, printed electronics, cyclic loading

## Abstract

For wearable electronic devices to be fully integrated into garments, without restricting or impeding movement, requires flexible and stretchable inks and coatings, which must have consistent performance and recover from mechanical strain. Combining Carbon Black (CB) and ammonia plasma functionalized Graphite Nanoplatelets (GNPs) in a Thermoplastic Polyurethane (TPU) resin created a conductive ink that could stretch to substrate failure (>300% nominal strain) and cyclic strains of up to 100% while maintaining an electrical network. This highly stretchable, conductive screen-printable ink was developed using relatively low-cost carbon materials and scalable processes making it a candidate for future wearable developments. The electromechanical performance of the carbon ink for wearable technology is compared to a screen-printable silver as a control. After initial plastic deformation and the alignment of the nano carbons in the matrix, the electrical performance was consistent under cycling to 100% nominal strain. Although the GNP flakes are pulled further apart a consistent, but less conductive path remains through the CB/TPU matrix. In contrast to the nano carbon ink, a more conductive ink made using silver flakes lost conductivity at 166% nominal strain falling short of the substrate failure strain. This was attributed to the failure of direct contact between the silver flakes.

## 1. Introduction and Literature Review

Wearable electronics, such as fitness trackers, are increasingly being used within the sport, fitness and health industries to improve our health, wellbeing and athletic performance [1]. However, many of these devices are based upon conventional electronics manufactured upon rigid silicone boards which can make garments uncomfortable and impractical for many uses [1,2,3]. For the large scale uptake of wearable e-textile technologies, devices must be lightweight, mechanically robust, durable, capable of withstanding bending and stretching, machine washable, aesthetically pleasing and must not impede the garment’s ability to conform to body curvatures [3,4,5,6]. From a sporting and fitness perspective it is also important that these devices do not impede sporting technique and performance. Integrated devices must be able to stretch to facilitate flexibility and to improve the conformability [7]. In textile applications about 15–20% strain occurs through the life of the product [1], therefore, this should be considered a minimum target for the extension capability of wearable devices. For devices to be truly anatomically compliant requires the inks to maintain a conductive path under severe mechanical deformations [4,6]. In humanoid robot application where large deformation is experienced such as the knee or elbow it is imperative the devices have stretchability [7]. Stretchable inks have been developed that can extend beyond 100% strain so this should be considered a target for stretchable inks [6]. Local strains induced during bending lead to tension and compression that can be very high, therefore, the ability to maintain an electrical network at high extensions is very important for wearable devices. For their uptake, printed electronics and their incorporation into garments requires robustness to last several cycles at large deformations (e.g., during creasing) together with moisture resistance from perspiration or washing cycles, to give wearables suitable lifetime.

Printing offers a high volume, scalable method to produce thin, flexible, stretchable and environmentally friendly devices produced by high throughput processes with reduced unit production costs [7,8,9]. Coating textiles with conductive materials has been identified as a way to add functionality, with recent applications including ultra-thin rechargeable batteries [10], healthcare monitoring and the measurement of health parameters [5,6,7], electronic entertainment devices [5], electroluminescent displays [11], and electrothermal heating [12]. Previously developed stretchable inks have been printed using techniques such as hand casting [13], mask printed [2], inkjet [6], stencil printing [14,15], screen printing [1,6,11], and flexography [6]. Inkjet printing requires inks that have well defined and narrow rheological parameters. This includes low viscosity and hence low particulate concentration to enable jetting and to prevent the nozzle clogging. This process facilitates relatively thin deposits of 1–10 μm at low speeds [6], restricting scalability and ability to produce thick film prints. Screen printing is a promising coating method for creating wearable devices as it allows for accurate patterning of thin films at economical rates onto a wide variety of substrate materials [16]. Many of these previously developed techniques have used low volume processing and lab-scaled techniques, therefore, there is a need for a scalable wearable ink, and to understand the effect that processing has upon its performance.

Conventional conductive inks have limited stretch-ability with poor durability to creasing and machine washing [3]. To achieve flexible and stretchable electronic devices with such inks requires the utilization of deformable structures. These allow the fabric to stretch and flex while limiting the strain on the ink, e.g., by using a zig-zag pattern that allows stretch in one direction only with minimal ink extension [4,7]. This limits the potential applications and design of the garments.

Alternatively, intrinsically flexible materials can be formulated by dispersing conductive fillers within an elastic polymer matrix, with electrical conductivity tuned by varying the loading of the filler material [4]. Conductive inks typically contain silver, carbon, gold or copper as the conductive element within a polymer/solvent blend. The choice and ratios of these materials is dependent on required conductivity, adhesion, ease of processing, economics and substrate flexibility [17]. In general, the electrical conductivity of the nano-composite inks will increase with filler loading. Unfortunately, elastic modulus, and therefore the stiffness of the material, also increases [7]. Metal based fillers are often used in conductive inks due to their relatively higher bulk electrical conductivity than alternatives such as carbon [7,9]. However, high loadings of metallic fillers are high cost, often make inks brittle and impair adhesion [3]. Stretchable inks consisting of silver flakes [2], Silver Nanowires [7], CNTs [6,14,15], and Ag/AgCl [14,15] have been developed that are capable of maintaining electrical conductivity at >100%, however, many of these materials are expensive or require extensive pre-treatment to enhance their dispersion. Intrinsically stretch-able materials, such as polymers (e.g., PEDOT: PSS20) and CNT compounds, generally permit high elongations but suffer from high resistance [1]. Therefore, there is a need for more conductive stretchable inks made from lower cost, more easily processable materials.

In addressing this, the advantages of carbon inks include their relatively low cost, disposability, ease of use, chemical inertness and their controllable electronic properties [4,18,19,20]. However, compared to their metallic counterparts the conductivity of non-metallic inks is relatively rather low [3]. Graphite is one of the widely used materials for stretchable and conformable electrodes, however, to achieve the required conductivity large amounts of filler are used, which also leads to the degradation of stretch-ability in graphite-based nanocomposites [7]. Thus, there is a trade-off between stretchability and conductivity. Graphite nanocomposites typically have weak mechanical strength as a result of poor interaction with the polymers [21]. Preparing high quality CNT and graphene inks for printed devices involves several challenges, such as their high cost [7,21], and their high surface area. Thus, obtaining inks with well-dispersed CNT/graphene suspension is difficult due to the strong van der Waals interaction [6,7,21]. Therefore, there is a need to develop stretchable carbon inks that are low cost, easily processable and readily dispersible. Pahaladegara et al. demonstrated stretchable inks made from CB that showed bulk resistivity of 71 Ω·cm and could be strained up to 50% [22], however, these levels of conductivity are too low to replace the role of conventional printed carbon inks.

Graphite nanoplatelets (GNPs) are a high aspect ratio graphitic nanocarbon, with a surface that can be functionalized to give improved interaction with polymers and solvents [23]. In a previous work by the authors Ammonia plasma functionalized GNPs were shown to be well dispersed within the TPU resin system, with ammonia plasma functionalized GNPs shown to have better dispersion stability, hypothesized to be a result of improved interaction between the particles and the TPU [24]. Plasma functionalization offers a dry, scalable, non-polluting, fast, one-step method of surface modification for carbon nanomaterials [25,26]. GNP only inks have been shown to have relatively low electrical conductivity with bulk resistivities of ~3 Ohm·cm [24]. Hybridizing graphitic material with carbon black (CB) has been shown to improve the electrical properties of inks and nanocomposites [13,16,18,27,28], as the submicron CB improves interparticle contact between neighboring graphitic particles. Michel et al., examined reduced GNP/PDMS inks and found at 20 wt% the composites had a bulk conductivity of 0.35 s/cm [29]. Quinsaat et al., examined the electro-mechanical performance of GNP/PDMS and GNP/CB/PDMS inks [13]. The GNP only inks showed low conductivity of 1.4 S/cm and 0.9 S/cm at 42 and 30 wt%, respectively, with the inks rupturing at relatively low strains of 18.5 and 30% nominal strain. At a 2:1 ratio of GNP:CB the GNP/CB/PDMS inks showed increased conductivity at 30 wt% to 7.8 S/cm, however, the ink was relatively brittle and barely survived 20% strains, making it unsuitable for wearable applications where larger strains may be encompassed. At 18 wt% the 2:1 GNP: CB GNP/CB/PDMS ink showed significantly lower conductivity at 0.25 S/cm, however, this ink was capable of surviving strains up to 140%. Given that the resistivity of conventional conductive carbon inks is in the range of 0.04 Ω·cm [18,20], for wearable inks to replace conventional inks in wearable devices a conductivity improvement is required. This demonstrates the need for a scalable, low-cost carbon stretchable ink, capable of withstanding large strains, while maintaining good electrical conductivity. Polymer selection is vitally important in designing a stretchable ink as it dictates film flexibility as well as its adhesive properties [30,31]. Elastomers such as Thermoplastic polyurethane (TPU) and polydimethylsiloxane (PDMS) have been filled with conductive particles to create intrinsically stretchable inks [1,2,3,13]. TPU was selected for this application as it offers a combination of favourable properties such as excellent elongation, high impact strength, good elasticity and biocompatibility [32]. Adhesion between the ink and the substrate is important for the mechanical properties of the finished device, thick ink films are often less flexible than the substrate it is printed on leading to poor adhesion [30]. Good adhesion has previously seen between a TPU based silver ink and TPU substrates [2]. TPU substrates have high abrasion resistance and high surface energy, giving significantly better adhesion to the conductive material than untreated PDMS [1].

For the ink to be scalable the solvent must not evaporate so quickly on the screen to cause drying in [30]. Previous studies have dissolved elastomers in harmful solvents such as toluene [13,29], however, health and safety considerations are of vital importance for effective scale up to larger quantities, therefore, the resin must form a stable solution in safe solvents.

Printed devices such as electrochemical sensors and batteries often consist of both carbon and silver inks [10,33]. The performance of the GNP/CB/TPU ink is compared to a stretchable silver in the same resin solvent system, to demonstrate the performance of the carbon against another conductive material in the same resin system. The effect of the ink’s rheological properties on the processability, print topography, conductivity and electromechanical properties is also examined. Previous studies have shown stretchable carbon inks made from expensive materials, using non-scalable processes, hazardous solvents and having relatively low conductivity. However, for the widescale uptake of wearable devices there is a requirement for the development of more highly conductive, stretchable inks, from lower cost, safe materials using scalable manufacture techniques. This study demonstrates the performance of a low cost, highly scalable, high conductivity stretchable screen-printable GNP/CB ink and compares it to a silver ink in the same resin system.

## 2. Materials and Methods

For the carbon composite ink, a blend of ammonia plasma functionalized Graphite Nanoplatelets (GNPs) and conductive Carbon Black (CB) was dispersed into a commercially available Thermoplastic Polyurethane (TPU) Resin in Diacetone alcohol. CB only [22] and GNP only [13,24,29] have been shown to have poor electrical performance, therefore as the aim of this study was to make a high conductivity, stretchable carbon ink. The ratio of GNP to CB was optimized for electrical conductivity with the most conductive GNP to CB ratio used in this study. To demonstrate the performance of the carbon ink its performance was compared with a silver ink comprising 55 wt% of 10 μm silver flake dispersed in the same TPU/solvent system. Silver flake was selected over spherical particles to give greater particle overlap and particle contact. This ink would also have a role in wearable technology as lower resistance printed circuit interconnects and busbars are required. Both inks were three-roll milled using the milling procedure set out in Phillips et al. [18], to further disperse the particles in the ink.

The stretchable inks were screen-printed using a semi-automatic, flat-bed, screen press (DEK, ASM Assembly Systems GmbH & Co KG, Munich, Germany) using a 54–70 polyester mesh, 2.5 mm snap-off, a polyurethane diamond edge squeegee 130 mm length with a 12 kg squeegee force and print/flood speeds of 70 mm/s onto a stretchable TPU substrate. The TPU substrate has a thickness of 80 ± 5 μm [34]. These inks were dried for 10 min at 70 °C and left for >24 h before testing to ensure all solvent had evaporated.

White light interferometry with 5× magnification lens and a 1.0 Field of view was used to assess the print thickness and roughness. Print thickness was taken by measuring the average difference in height between the substrate and the print at opposing edges of 5 of the prints. The average surface roughness (Sa), and the peak surface roughness (Sz), was taken from the centre of 8 of the prints, away from any edge effects.

Samples were cut from the center of the prints and a high-resolution Field Emission Gun Scanning Electron Microscope (FEG-SEM) (JEOL 7800F) (JEOL Ltd, Tokyo, Japan) was used to take images of the dried prints at magnifications of 1000, 10,000 and 27,000×. For the 1000× carbon print a 10 kV lower electron detector was used, while for the 27,000× carbon print and the 1000, and 10,000× silver prints an 8 kV upper electron detector was used. ImageJ software (ImageJ 1.52a, National Institutes of Health, Bethesda, MD, USA) was then used to analyse the particle sizes, by taking 10 calibrated measurements and presenting the results as the mean ± the standard deviation.

The sheet resistance of the prints was measured using a 4-point probe (SDKR-13, NAGY Messsysteme GmbH, Gäufelden, Germany) with a 1.3 mm gap with a digital multi-meter (Keithley, Tektronix, Beaverton, OR, US). The sheet resistance of the carbon ink was taken from 5 measurements along the centre of 8 of the block carbon prints, using a correction factor of 4.5129 as proposed by Smits [35]. The sheet resistance of the silver ink was taken from 5 measurements from 6 prints along the centre of the 0.5 × 15 cm prints, using a correction factor of 3.2248, see Smits [35]. Bulk resistivity was then calculated by multiplying the sheet resistance in Ohms by the print thickness in cm for 6 samples. Bulk conductivity was calculated as the inverse of bulk resistivity.

The rheological properties of the ink were measured using a stress-controlled rheometer, Malvern Kinexus Pro (NETZSCH-Gerätebau GmbH, Selb, Germany) A 40 mm diameter roughened parallel plate geometry was used to negate the risk of wall slip due to the heavily filled nature of the fluids. The temperature was kept constant at 25 °C using a Peltier plate system. The inks were first ramped to 100 s^−1^ to ensure consistent pre-shear throughout the samples, as it was hypothesised this shear rate would be greater than any applied during the application and stirring of the sample. Shear viscosity values were taken from the down shear ramp from 100 to 0.1 s^−1^. The experimental error on the Malvern Kinexus was examined to be <0.13 Pa·s at 1 s^−1^ across three measurements [24].

Procedures compliant with International Standards were used for the determination of tensile properties of plastics, BS EN ISO 527-1:2012, and films and sheets, BS EN ISO 527-3:1996, were used to guide the measurement of the tensile properties of the coatings [36,37].

Three 15 × 0.5 cm test strips were cut from the prints using a sharp knife, with the edges of the samples inspected for any notches or tears.

Adhesive copper tape was attached 0.5 cm from opposing ends of the test sample to leave a 14 cm electrical path length. Clips were then attached to ensure good electrical contact between the wires and the flat printed samples. Resistances were measured continuously during tensile testing using a source meter. Tensile testing was performed using a Hounsfield Tensile Tester (Hounsfield, Tinius Olsen TMC, Horsham, PA, US) with a 100 N load cell, gripped to give a test section length of 10 cm and an extension speed of 50 mm/min (Figure 1a).

The samples were tested in a controlled lab environment (18 ± 1 °C, 50 ± 10% relative humidity). The nominal strain used for testing was calculated from the measured gripper displacement. Three samples were tested to extension at break. A further three samples were cycled to 10% nominal strain 30 times. Compression tests were also performed using a 10 kN load cell and a 250 N compressive force on 3 further printed samples (Figure 1b). The samples were compressed with the ink on the outside radius initiating large tensile strains, to simulate the inks being creased during use.

Formulas:

Engineering Stress:(1)$$\sigma =\frac{F}{A}$$
where;

*σ* is the stress value in question, expressed in megapascals (MPa);

*F* is the measured force concerned, expressed in newtons (N);

*A* is the initial cross-sectional area of the specimen, expressed in square millimetres (mm^2^).

Nominal Strain Percent:(2)$$\varepsilon_{t}=\frac{L_{t}}{L}\times 100$$
where;

*ε_t_* is the nominal strain, expressed as a percentage;

*L* is the gripping distance, expressed in millimetres (mm)

*L_t_* is the increase in the gripping distance occurring from the beginning of the test, expressed in millimetres (mm).

Change in Resistance:(3)$$Change\;in\;Resistance=\frac{Change\;in\;Resistance}{Initial\;Resistance}\times 100$$

## 3. Results

Prints were performed as set out in the Materials and Methods section and to facilitate comparison, the carbon and silver inks have been printed at similar thicknesses of 7.96 and 8.73 μm, respectively. The performance of the two inks can be seen in Table 1 and these have been derived from repeat measurements as set out in the table. The sheet resistance of the stretchable GNP/CB/TPU ink was 230 Ω/□ while the silver ink had significantly lower sheet resistance at 0.078 Ω/□. The sheet resistance of the silver ink is considerably lower than the carbon ink owing to the silver’s higher bulk electrical properties.

Sheet resistance is a function of layer uniformity and thickness; therefore, bulk resistivity is a more appropriate method of comparing the electrical properties of inks in the literature as it accounts for the thickness. The bulk resistivity of the GNP/CB/TPU ink was calculated to be 0.196 ± 0.013 Ω·cm, with the bulk conductivity therefore 5.097 ± 0.367 S/cm. The bulk resistivity of the GNP/CB/TPU carbon ink is significantly lower than the low-cost carbon inks in the literature such as 71 ± 4 Ω∙cm CB/TPU ink used by Pahalagedara et al. [22], the 0.25 S/cm found for the 18 wt% GNP/CB/PDMS screen printed by Quinsaat et al. [13], and is significantly closer to the the 0.04 Ω∙cm found for traditional non-stretchable, conductive carbons such as Graphite/CB/Vinyl inks used by Phillips et al. [18], and the 0.038 of Potts et al. [20], That the GNP/CB/TPU ink showed similar performance to the commercially available carbon inks for printing indicating the GNP/CB/TPU ink could be a promising stretchable alternative conductive carbon ink.

The roughness of the surface of a printed feature is a consequence of the substrate topography, the topography of the printed layer, which is a function of the print process and the ink’s rheological properties, and roughness due to particles [10,11,12,16,17,18]. When the carbon coating was printed onto the substrate the surface roughness approximately halved compared to the TPU substrate, reducing it from 1.28 ± 0.03 μm (s.d) to 0.67 ± 0.04 μm (s.d), as the ink spread to fill the valleys in the substrate (Figure 2). With an Sa of 0.67 μm, the GNP/CB/TPU ink has a significantly lower roughness than the 1.7 μm found for a 30 wt% Graphite/CB ink by Potts et al. [20], and the ~1 μm found for a 21.7 wt% 1.8:1 Graphite/CB ink found by Phillips et al. [18].

To gain an understanding of the above, screen-printing is a process in which ink is forced through the open areas of a stencil supported by a mesh of synthetic fabric supported by a frame, and onto the substrate underneath by drawing a squeegee over the stencil [30]. The flow of ink through the mesh is significant since it determines the uniformity of the printed surface which is an important factor in conductive circuits [9]. Furthermore, for high quality screen prints, characterised by dimension and surface topography, it is well known that inks need to show a shear thinning characteristic and consequently ink rheology was tested as part of the development.

Material rheology impacts processes subsequent to ink transfer [16]. The more viscous an ink the more difficult it is for the ink to spread into an even film [31]. Higher low shear viscosities and steeper shear profiles give higher print roughness owing of the inability of the ink to level following release from the screen [18], leaving a relatively rough surface with mesh marking [20]. The rheological properties of the two inks and the unfilled resin can be seen in Figure 3. The unfilled TPU resin has a relatively low viscosity at 2.46 Pa·s at 1.18 s^−1^ and shows approximately Newtonian behaviour. The addition of particles to the unfilled TPU resin increased the viscosity especially at low shear-rates, with suspensions showing the desired non-Newtonian behaviour over the whole measurement range. The silver had the greatest effect on the low shear viscosity and at 3 s^−1^ falls below that of the carbon where the shear forces break down any interparticle interactions reducing their effect on viscosity. The addition of the GNP and CB to the unfilled TPU resin has a similar benefit in giving the ink shear thinning properties. In combination, they form a network. However, this network is weaker, reflected in a smaller shear thinning effect.

The silver prints were significantly rougher than the carbon with average surface roughness of 1.19 µm. The white light images show a structure of peaks and valleys within the topography of the prints, with the difference in the thickness of the coatings in these areas as high as 5.33 µm (Figure 2b). The high low-shear viscosity of the silver ink prevents the ink from levelling having passed through the screen to leave a relatively rough surface with mesh-marking. In contrast the relatively low surface roughness of the carbon prints suggests the hybrid GNP/CB was able to flow freely through the screen and level following the screen pulling away from the substrate to create a smooth, uniform coating (Figure 2a).

A scanning electron microscope was used to examine the microstructure of the coatings (Figure 4). Figure 4a shows a lower resolution image of the GNP/CB composite which exhibits a near homogeneous surface, with the GNP flake size shown to be 4.05 ± 0.89 μm. To gain further insight, Figure 4c shows a typical high-resolution image that captures the CB particles and the GNP platelets. The CB can be identified clearly as its scale is typically 0.05 ± 0.02 µm in comparison with the GNP that is typically 4.05 ± 0.89 µm. The smaller, spherical CB particles appear to decorate the face of the platelets, to form a dense 3D-network. Indeed, despite exposure to high levels of strain rate during the ink making and printing processes the sub-micron CB particles appear to have largely attached in groups to the face of the GNP platelets. Measurement of the visible GNP platelet edges in Figure 4c, shows the plate thickness to be 0.010 ± 0.003 μm, which given the previously measured thickness of 4.05 ± 0.89 μm, demonstrates the high aspect ratio of the GNPs. The corresponding images for the silver ink are shown in Figure 4b,d. In contrast the smaller silver flakes appear to form a less homogenous microstructure, with voids present in the structure (Figure 4b). The higher resolution image in Figure 4d shows clearly that adjacent flakes leave small voids in the ink film as the flakes tend to lock together rather than flow that is typical of spherical particle geometries.

The mechanical properties of the coated samples were dominated by the substrate because of the substantially higher thickness of the substrate of 80 µm, compared to the approximately 8 µm thickness of the printed layer (Figure 5a). Engineering stress and nominal strain were used to characterise the mechanical properties as the samples were drawn to failure. The uncoated TPU substrate shows a typical soft rubber-like material stress–strain response with a small linear elastic region at nominal strains of approximately <5%, followed by a plateau before the substrates break at 330% nominal strain. The carbon coating increased the shear stress at nominal strains <50% but decreases the strain at break to 305% nominal strain. This increased stress at low nominal strains could be a result of the carbon coating stiffening the substrate, giving a reinforcing effect.

The difference in extension at break is within the standard deviation of the substrate therefore the sample break is probably the result of substrate failure.

The silver ink showed a similar stress–strain curve to the uncoated substrate indicating the silver coating does not have the same reinforcing effect as the carbon.

Change in resistance with nominal strain allows for a better comparison of the two inks as carbon and silver have significantly different bulk electrical properties (Figure 5b). The change in resistance with applied strain is higher in the silver ink, however, the inks show a similar shape response until at above 100% nominal strain the silver prints show a rapid increase in resistance losing electrical conductivity completely at 166%, while the carbon maintains a conductive network until 305%. Although having a higher initial resistance the carbon inks maintain electrical conductivity up to the point of failure at a nominal strain of 305% (Figure 5b). The silver ink has a lower initial resistance however, it loses electrical conductivity at a significantly lower nominal strain of 166% (Figure 5b) indicating the conductive network has already been broken despite the coating continuing to be extended to mechanical break at nominal strains 346% (Figure 5a).

Having explored the tensile failure, the prints were strained cyclically to 10% nominal strain to explore behaviour at levels close to the requirements for normal wearable textile applications. The strain was chosen to just exceed the elastic range of the materials. The carbon ink was also stretched repeatedly to 100% nominal strain to examine the cyclic response at extreme strains.

The electro-mechanical response of the carbon and silver inks when cyclically loaded to 10% nominal strain can be seen in Figure 6. During the first cycle the resistance of the carbon ink increases by 103% at 10% nominal strain, with the resistance returning to 65% of its maximum value once the nominal strain returns to 0%. The resistance of the carbon ink continues to recover to 53% of its original value at 4.3% nominal strain in the next cycle. The increase in resistance at 0% nominal strain is likely a result of the permanent increase in the electronic pathlength as a result of permanent deformation to both the TPU substrate and the ink, this permanent increase in the length of the coating would be expected as 10% nominal strain is outside of the linear elastic region of the coatings (Figure 5a). The re-orientation of the particles in the ink following initial deformation has previously been suggested as a mechanism behind this increased resistance following initial strain [38]. Having been strained to 10% nominal strain the electrical resistance of the carbon inks continued to recover following the sample returning to 0% nominal strain, up till approximately 5% strain in the next cycle. This is as the sample continues to recover while the sample is slack in the grips following its permanent deformation. The coating and substrate have experienced a permanent deformation of approximately 4%, increasing the electronic path length.

There are smaller increases in the resistance of the carbon print with increasing number of cycles at 10% nominal strain, decreasing from 103% in the first cycle before becoming consistent at an increase in resistance of 80.6% between cycles 15–20. The recovered strain following each cycle becomes consistent at to 51.3% during cycles 15–19. The decrease in the change in resistance and the increase in the consistency of the change in resistance to 10% nominal strain with increasing number of cycles suggests a permanent change in the microstructure of the printed layer, possibly orienting the particles in the direction of the strain to better facilitate strain in this direction.

The change in resistance of the silver prints at 10% nominal strain increases with the number of cycles (Figure 6), with the strain at 10% nominal strain increasing from 129% in the first cycle to 267% at cycle 20. This indicates there is a change in the microstructure of the silver prints with every strain, which is not fully recovered when the strain is removed and negatively affects the electronic properties of the print. This result renders the silver ink formulation unsuitable for large strain applications, pointing to the need for the exploration of new formulations.

The effect of cyclically stretching to 100% nominal strain on the electrical properties of the carbon inks can be seen in Figure 7. The resistance of the prints at 0% strain increases after the first cycle as there is a permanent increase in the electronic path length owing to the permanent increase in the length of the print and the substrate. The change in resistance at 100% nominal strain decreases with the number of cycles from 6317% after the first cycle, to 5512% after the 20th cycle, again indicating the change in the microstructure of the prints as the conductive materials become aligned with the strain to stabilise their resistance excursions to cyclic strain.

### Results Summary

The carbon ink-maintained conductivity to >300% nominal strain. During cyclic strains to 100% nominal strain, following an initial increase in resistance, the increase in resistance at 100% nominal strain decreased with number of cycles. The microstructure of the carbon is changing to accommodate the high strains, reducing the effect of strains on the electrical properties. In contrast the silver ink lost electrical conductivity at 166%. When straining to 10% nominal strain the resistance of the silver increased with the number of cycles.

On a micro level, the smaller, lower aspect ratio silvers lie largely parallel to the substrate with large voids present between neighbouring particles (Figure 4b,d). Araki et al. showed that as mechanical strain increases the silver particles are pulled further apart from one another decreasing the number of electrical contacts between neighbouring flakes and subsequently increasing the resistance [2]. The application of a large strain pulls these particles further away from one another, reducing interparticle contact until eventually at 166% a conductive pathway through the print is no longer available. The increases in resistance become greater with the number of cycles could indicate that these particles are being moved to new positions within the coating with every cycle and this is not recovered when the strain is removed. This leads to cumulative increases in the resistance with the number of cycles.

The surface of the GNP flakes was decorated with CB, giving a dense 3D network of particles, with the high aspect ratio platelets aligned at various angles to the substrate (Figure 4a,c). The application of strain would be expected to increase the distance between neighbouring GNPs in the same manner as the silver particles, decreasing interparticle contact and therefore increasing resistance. However, this strain could also align the platelets in the direction of strain, parallel to the substrate, allowing them to maintain an electrical contact even at high strains. This permanent change in the microstructure would result in a decreased effect of strain on the electrical properties while the sample is under tensile strain.

On a macro level the relatively low viscosity of the carbon ink allowed the ink to easily pass through the screen to create a print of low surface roughness, Sa = 0.67 μm. The ability of the ink to flow out under the mesh to form a consistent film is important for the good adhesion between the ink and the substrate [30]. In contrast the high, low-shear viscosity of the silver ink meant that it was unable to relax having passed through the screen and formed a rough surface, with Sa = 1.19 μm related to variances in print thickness of <5 μm (Figure 2c,d). Under tensile strains, these mounds of conductive material could be pulled away from each other, with higher stresses being experienced by the thinner areas between the mounds, until the conductive material was pulled far enough away from one another that the conductive network becomes broken.

The mechanical properties of the prints were dominated by the substantially thicker substrate. Good adhesion between the ink and the substrate are likely to play an important role to help facilitate the transfer of strain from the substrate to the printed sample. That the carbon ink maintained a conductive network up until substrate break suggests good adhesion between the ink and the substrate.

Strains locally during bending and compression can be very high, therefore a test was developed using a force of −247 ± 35 N to compress a crease into the sample with the resistance change measured. The printed side of the sample was on the outside of the compression to maximise strains. During compression, the resistance increased by 3.87% ± 0.06 (s.d) and following the removal of this compressive force the resistance of the sample instantly recovers to a 1.83% ± 1.00 (s.d) increased resistance. This permanent change in the resistance is likely a result of an increase in the electronic path length, visible as a crease in the sample. The change in resistance is significantly lower than for the tensile testing, as the stresses applied during compression are more local than those during tensile testing.

## 4. Conclusions

A novel GNP/CB/TPU screen-printing ink has been developed using low-cost carbon materials and non-harmful solvents making it a viable candidate for scale up. The ink also showed good print performance on a semi-automatic screen press, with the ink freely passing through the screen to create a uniform coating with surface roughness Sa = 0.67 μm.

To replace conventional inks for use in wearable electronics new stretchable inks must first develop competitive electrical properties relative to conventional inks. The GNP/CB/TPU ink showed improved electrical performance when compared with similar inks in the literature [13,22,29] with a bulk resistivity of 0.196 ± 0.013 Ω·cm, which is close to the performance of conventional conductive inks [18,20]. A novel GNP/CB/TPU ink has been developed that could maintain electrical conductivity up to 300% nominal strain, cyclic straining to 100% and showed good resistance to compression.

The mechanical properties of the coated samples were dominated by the substantially higher thickness of the substrates compared to the prints. The performance of the GNP/CB/TPU ink was compared to a silver ink within the same resin. The carbon ink maintained electrical conductivity until sample break at approximately 300% nominal strain whereas the Silver flake coating lost electrical conductivity at around 150%. The carbon ink also showed increased resistance to repeated cyclic straining. During cyclic strain testing to 10% after an initial permanent deformation during the first strain cycle the carbon ink showed consistent electrical response to applied strain, whereas the resistance of the silver ink increased with every cycle as the sample continued to undertake permanent deformation.

## Figures and Tables

**Figure 1 nanomaterials-11-01200-f001:**
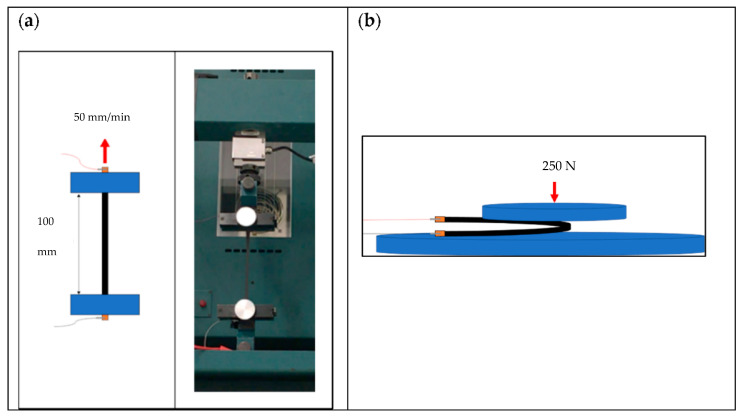
Electromechanical testing of the stretchable inks. (**a**) Simultaneous measurement of electrical and extensional tensile properties (**b**) Simultaneous compression and measurement of electrical properties.

**Figure 2 nanomaterials-11-01200-f002:**
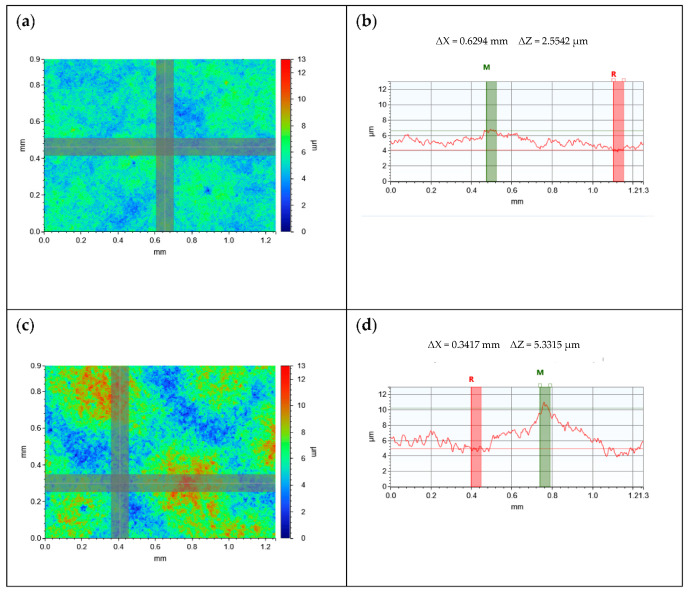
White light images at 5× magnification of the centre of single layer prints of carbon (**a**) and (**b**) the, silver (**c**) and (**d**). These respectively represent a 1.2 mm × 0.9 mm scan area, where wrl represents the height of the surface in μm and cross-sectional X-profile of the white light images showing the roughness across the surface of the prints in microns, where ΔZ represents the difference in height between the reference area (R) and the measure area (M).

**Figure 3 nanomaterials-11-01200-f003:**
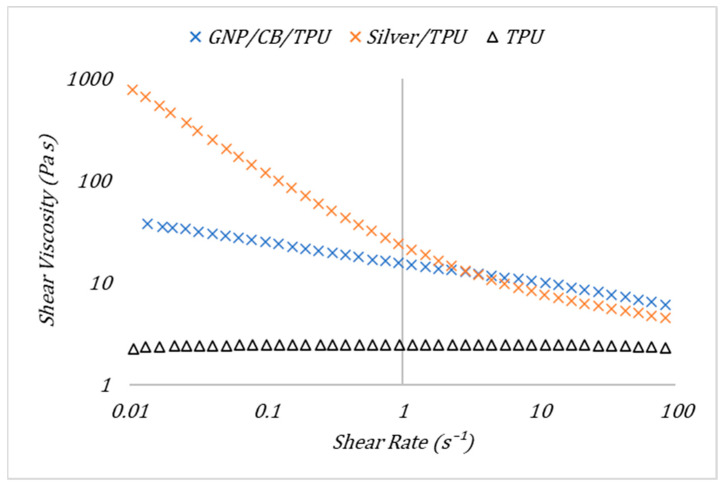
The effect of shear rate on the shear viscosity of the GNP/CB/TPU ink, the Silver/TPU ink and the unfilled TPU resin system from a downwards shear ramp.

**Figure 4 nanomaterials-11-01200-f004:**
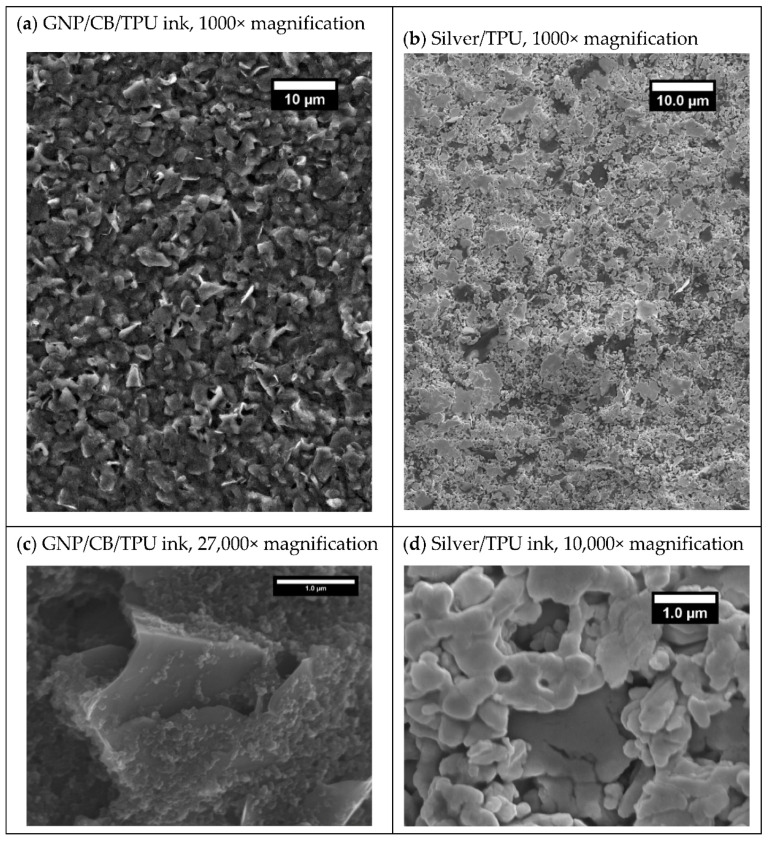
Scanning electron microscope images of (**a**) GNP/CB ink at 1000×, (**b**) Silver flake ink at 1000× (**c**) GNP/CB ink at 27,000×, and and (**d**) Silver flake ink 10,000× magnification.

**Figure 5 nanomaterials-11-01200-f005:**
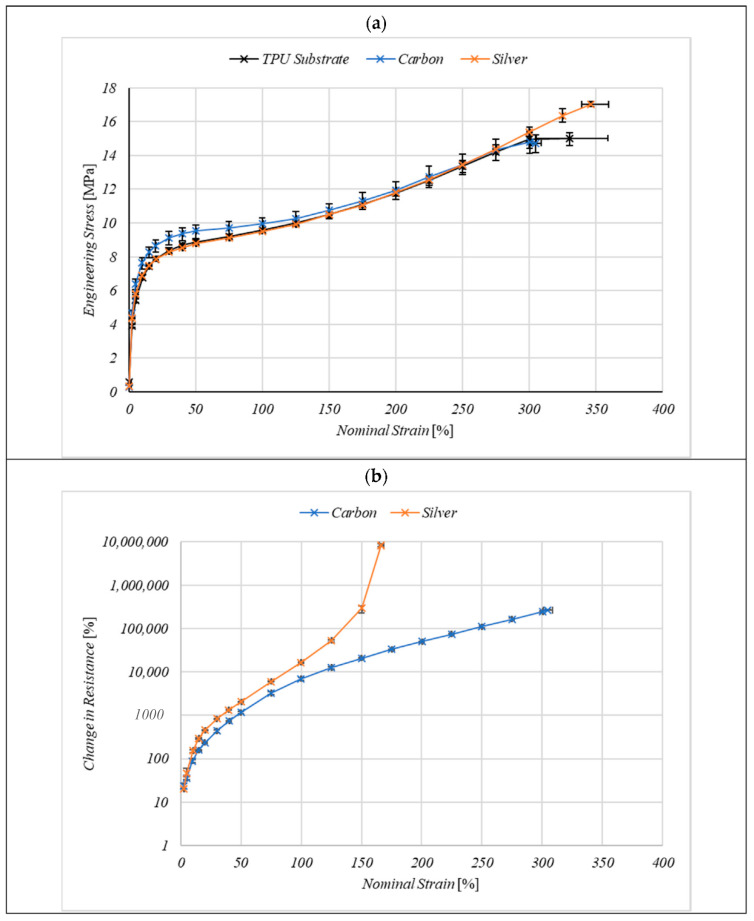
(**a**) The effect of carbon and silver ink coatings on the stress–strain curve of the TPU substrate during a maximum extension test at 50 mm/min. (**b**) the effect of nominal strain on the electrical resistance of the coatings. Data represent the mean of *n* = 3 samples where error bars represent the range.

**Figure 6 nanomaterials-11-01200-f006:**
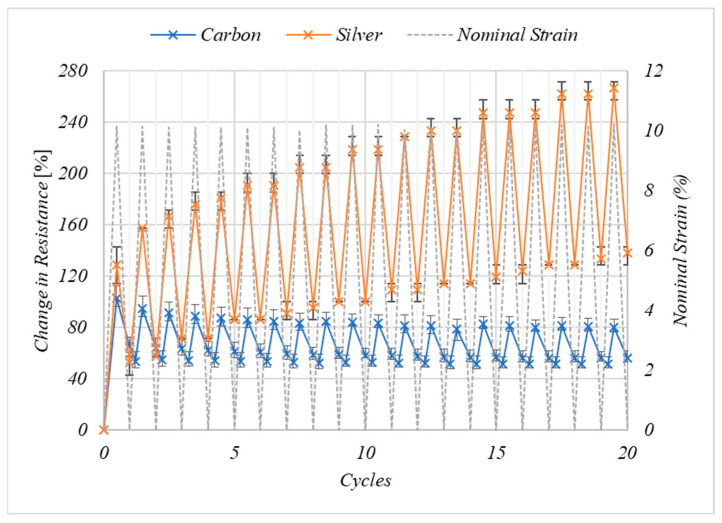
The effect of 20 cyclic strains to 10% nominal strain on the resistance of the coatings relative to their value at 0% nominal strain from *n* = 3 samples where error bars represent the range of values.

**Figure 7 nanomaterials-11-01200-f007:**
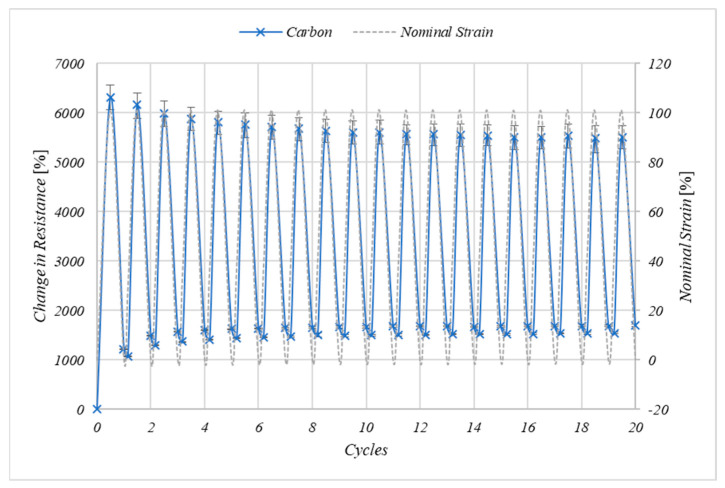
The effect of cyclic straining of the carbon ink to 100% nominal strain at 50 mm/min on the electrical resistance of the *n* = 3 samples where error bars represent the range of values.

**Table 1 nanomaterials-11-01200-t001:** The printed properties of the Carbon and Silver print. Printed thickness is taken from *n* = 10 measurements and shown ±s.d. Average surface roughness (Sa) and Peak-peak surface roughness (Sz) is shown from *n* = 8 measurements and shown ±s.d. Sheet resistance was taken 40 measurements from *n* = 8 samples for the carbon ink and 30 measurements from *n* = 6 samples for the silver and shown ±s.d.

Ink	Printed Thickness [μm]	Sa [μm]	Sz [μm]	Sheet Resistance [Ω/□]
GNP/CB	7.96 ± 0.78	0.67 ± 0.04	10.24 ± 1.67	230.102 ± 12.40
Silver	8.73 ± 0.51	1.19 ± 0.08	11.91 ± 1.22	0.078 ± 0.0005

## Data Availability

Data is contained within the article.

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
