# Peer review of "Stretchable Carbon and Silver Inks for Wearable Applications"

_nanomaterials, 2021, doi:10.3390/nano11051200_

Round 1
Reviewer 1 Report
In my opinion, the article probes an interesting problem. Nevertheless, before I recommend this work for publication, the authors should carefully and fully address my comments shown below:
(1) The authors need to provide more details on the mathematical formulation and how these equations have been derived in a more systematic fashion.
(2) What are the main assumptions made to solve the equations? The authors should include all the assumptions made in the main text as it helps other researchers in the field to advance this work.
(3) The authors need to highlight the importance of this work and how this is different from other studies. Consider citing few articles published in this journal, Nanomaterials.
(4) Figure 2 description is not enough. Each part must be explained separately. ".... the ink’s rheological properties, roughness due to particles" why?
(5) The result of the effect of carbon and silver ink coatings on the stress-strain curve at the range of 0 to 300% are very close. Do these tests have reproducibility? What is the reason?
(6) The conclusions section is very long and general.
Author Response
In my opinion, the article probes an interesting problem. Nevertheless, before I recommend this work for publication, the authors should carefully and fully address my comments shown below:
(1) The authors need to provide more details on the mathematical formulation and how these equations have been derived in a more systematic fashion.
Equations have been added to the methods sections
(2) What are the main assumptions made to solve the equations? The authors should include all the assumptions made in the main text as it helps other researchers in the field to advance this work.
Equations used to calculate and solve these equations have been added to the methods section
(3) The authors need to highlight the importance of this work and how this is different from other studies. Consider citing few articles published in this journal, Nanomaterials.
The abstract, introduction and conclusions have been re-written to re-focus the work to emphasize the novelty. The authors feel the novelty is now better summed up at the end of the introduction by the following. Previous studies have shown stretchable carbon inks made from expensive materials, using non-scalable processes, hazardous solvents and low conductivity. However, widescale uptake of these devices will require the development of more highly conductive, stretchable inks, from lower cost materials using scalable manufacture techniques. This study demonstrates the performance of a low cost, highly scalable, high conductivity stretchable screen-printable GNP/CB ink and compares it to a silver ink in the same resin system.
There is also a focus on the processing side and how the ink rheological properties and processability may affect final stretch performance
More papers from nanomaterials have been referenced to help with this.
(4) Figure 2 description is not enough. Each part must be explained separately. ".... the ink’s rheological properties, roughness due to particles" why?
Rheological data of the inks has been added to help explain how the inks rheological properties affect the way in which the ink passes through the screen and subsequent print topography.
(5) The result of the effect of carbon and silver ink coatings on the stress-strain curve at the range of 0 to 300% are very close. Do these tests have reproducibility? What is the reason?
The stress-strain curve for the carbon ink, silver ink and the substrate are very similar as the mechanical properties of the substrate dominate owing to it’s much greater thickness of 80μm compared to the 8μm of the inks. We believe these tests to be repeatable.
(6) The conclusions section is very long and general
The conclusions have been shortened to make it more concise, we feel this also helps to bring out the novelty of the work.
Reviewer 2 Report
This work by Claypole et al. l compares the performance between carbon and silver stretchable electronics. The materials are embedded in a polyutherane stretchable matrix, and the conductivity persists even under cyclic mechanical straining. Experimental results back the claims and the physical properties are properly characterized. I don’t see any bias from the authors regarding citation. While the work can be valuable in some respects, the paper does not highlight the scientific contribution and in general, seem like a derivate study of state of the art in stretchable printable electronics, eg Advanced Electronic Materials, 2017, 3(1), p.1600260. and the author’s previous work. This paper really needs to indicate more clearly its novelty.
I would support publication after the following minor comments.
- What is the real innovation here?, the materials or fabrication method doesn’t seem to be better or cheaper than those presented in the literature for the last 5 years.
- Can the authors describe how they selected the ratio between Graphite 98 Nanoplatelets and Carbon Black?, as a minor suggestion, the same test consisting of each of the materials could help understand the contribution of each material
- Figure 7 sketch seems a little misleading, as they look like carbon nanotubes instead of particles.
Author Response
Comments and Suggestions for Authors
This work by Claypole et al. l compares the performance between carbon and silver stretchable electronics. The materials are embedded in a polyurethane stretchable matrix, and the conductivity persists even under cyclic mechanical straining. Experimental results back the claims and the physical properties are properly characterized. I don’t see any bias from the authors regarding citation. While the work can be valuable in some respects, the paper does not highlight the scientific contribution and in general, seem like a derivate study of state of the art in stretchable printable electronics, eg Advanced Electronic Materials, 2017, 3(1), p.1600260. and the author’s previous work. This paper really needs to indicate more clearly its novelty.
I would support publication after the following minor comments.
- What is the real innovation here?, the materials or fabrication method doesn’t seem to be better or cheaper than those presented in the literature for the last 5 years.
The abstract, introduction and conclusions have been re-written to re-focus the work to emphasize the novelty
The effect of the inks rheological properties on the processability, print topography, conductivity and electromechanical properties is examined. Previous studies have shown stretchable carbon inks made from expensive materials, using non-scalable processes, hazardous solvents and low conductivity. However, widescale uptake of these devices will require the development of more highly conductive, stretchable inks, from lower cost, safe materials using scalable manufacture techniques. This study demonstrates the performance of a low cost, highly scalable, high conductivity stretchable screen-printable GNP/CB ink and compares it to a silver ink in the same resin system.
There is also a focus on the processing side and how the ink rheological properties and processability may affect final stretch performance
This paper has been about the development of a new, low cost, highly scalable, high conductivity stretchable ink and we have re-written to help focus on that.
- Can the authors describe how they selected the ratio between Graphite 98 Nanoplatelets and Carbon Black?, as a minor suggestion, the same test consisting of each of the materials could help understand the contribution of each material
The ratio of graphite nanoplatelets and carbon black was optimised in an earlier study for electrical conductivity. Carbon black only and GNP only inks have been shown to have poor electrical conductivity relative to conventional carbon printed electronics inks. Some more information has been added into the introduction and the methods to help explain why we used a graphite nanoplatelet GNP hybrid.
- Figure 7 sketch seems a little misleading, as they look like carbon nanotubes instead of particles.
Figure 7 has been removed as we felt it was a bit misleading and did not add to the understanding described within the text.
Round 2
Reviewer 1 Report
The revised manuscript is significantly improved. However, because some parts are still obscure, the manuscript content is still not a quality fit for publication. I, therefore, recommend a new revision of the manuscript to bring its content up to a suitable level.
Almost half of the article has been revised and changed. Section SEM is still not well explained. the authors claim that the sub-micron CB particles have largely attached in groups to the face of the particles. How can this be scientifically proven? The authors should explain the results obtained from the SEM analysis fig 4 b and d more thoroughly. Also, the authors should indicate experimental errors throughout the paper. What is the authors' goal in adding Figure 3 to the new version?
Author Response
All the issues have been addressed raised by the reviewer. We have left in track changes for some of the alterations, but only picked up on the editors request to include track changes on the last round of changes to the paper as it was passed back and fore between myself and the other senior academic on the paper.

Round 3
Reviewer 1 Report
The article can be accepted for publication.